# Abortion and contraception for incarcerated people: A scoping review

**Martha Paynter** [1]*, **Paula Pinzón Hernández**[2], **Clare Heggie**[3], **Shelley McKibbon**[4], **Sarah Munro**[2]

1 Faculty of Nursing, University of New Brunswick, Fredericton, New Brunswick, Canada, 2 Department of Obstetrics and Gynaecology, The University of British Columbia, Vancouver, British Columbia, Canada, 3 IWK Health Centre, Halifax, Nova Scotia, Canada, 4 WK Kellogg Health Sciences Library, Dalhousie University Libraries, Halifax, Nova Scotia, Canada

* martha.paynter@unb.ca

## Abstract

### Background

Women experiencing incarceration have higher rates of unmet contraceptive needs and rates of abortion than the public. Incarceration presents multiple potential barriers to accessing abortion and contraception care, including prison security protocols, prison locations, lack of access to care providers, stigma, and low health literacy. The objective of this scoping review is to understand the extent and type of evidence in relation to contraception and abortion access for people experiencing criminalization and incarceration.

### Methods

We used the Joanna Briggs Institute methodology for scoping reviews and include empirical research with people experiencing criminalization or incarceration and/or with prison staff; with respect to prescription contraception or abortion access, while in custody or after having experienced incarceration/criminalization. Databases searched include CINAHL, APA PsycInfo, Gender Studies, Medline (Ovid), Embase, Sociological Abstracts, and Social Services Abstracts. The search yielded 6096 titles of which 43 were included in the review.

### Results

Our search yielded 43 studies published between 2001 and 2021 across six countries. The studies included qualitative, quantitative, and mixed methods designs. The main outcomes of interest included contraceptive use; attitudes towards abortion, contraception, and pregnancy; and barriers to care. Barriers identified included lack of onsite access to options, contraceptive coercion by providers, financial costs, and disruptions to medical coverage and insurance status which incarcerated.

### Discussion

Evidence indicates that people in prison face significant barriers to maintaining continuity of contraceptive methods, abortion access, and reproductive health guidance. Some studies articulated participants felt judged when discussing contraception with prison-based health

**Data Availability Statement:** All data are present within the paper and its Supporting information files.

**Funding:** The Contraception and Abortion Research team at the University of British

Columbia, awarded to MP, provided support for this project.

**Competing interests:** The authors have declared that no competing interests exist.

care providers. Geographic location, out-of-pocket payments, and trust in health care providers were reported as barriers to access.

## Conclusion

Incarceration presents considerable challenges to the access of contraception and abortion care. Future research should examine the interaction between institutional security policies and procedures on care seeking, the experiences of underserved and hyper-incarcerated groups, and the impact of being denied access to contraception and abortion and experiences of criminalization.

## Introduction

Barriers to pregnancy prevention and termination result in gendered social, economic, and political inequality and increase risk of intimate partner violence. Intersecting with the dangerous potential consequences of criminalizing access to abortion are the reproductive health harms associated with incarceration [1]. Incarcerated people experience logistical, financial, and geographic barriers to care, such as distance from services; restrictive security, transportation, and escort policies and practices; staff shortages; and prohibitive private costs. They also face stigma and threats to privacy and confidentiality from both institutional health care providers and correctional staff [2], and barriers to health information and literacy [3]. Even where abortion services are completely decriminalized, lack of understanding about how to access care is a serious impediment to seeking services. Further, one of the most significant improvements to access in decades–the increasing availability of mifepristone and misoprostol medications for home abortion–is problematic in prison environments that involve heavy surveillance and lack of access to basic self-care supplies.

There is very little information available internationally about the rate of abortion or contraception use among incarcerated people. Recent US data suggests approximately 1% of pregnancies among incarcerated women resulted in abortion [4], a rate far lower than that among the public. Researchers have found people who experience incarceration have high rates of unplanned pregnancy and higher than average rates of fertility [5], potentially generating high and unmet need for abortion. The recent reports of low abortion rates among incarcerated women in the US suggest this trend cannot be explained by demographics alone and may be due to unwarranted variation in access to care for this population.

The relationship between incarceration and access to abortion is poorly understood. A systematic review of contraception needs and available services among incarcerated women in the United States completed in 2020 identified 25 studies on the subject [6]. Results of the synthesis indicated incarcerated women desire access to contraception from carceral health care systems, but face barriers including lack of provider training about birth control methods as well as concerns about the ability to continue/discontinue their chosen method in community because of cost and access to providers. However, the review did not address abortion, nor did it investigate experiences outside of the US–a unique setting with respect to health and prison systems. The objective of our review is to assess the extent of the literature on both abortion and contraception for people in prison internationally. This review asks what is known about access to abortion and contraception among people experiencing criminalization and incarceration.

## Methods

In this review we used the Joanna Briggs Institute (JBI) Methodology for scoping reviews [7]. The populations of interest included women, trans, and non-binary people who have experienced arrest, criminalization, or incarceration, or people in administration roles in prisons designated for women who can speak to policies and practices in those settings. The concept of interest was access to, knowledge about, and use of abortion and prescription contraception. The context was criminalization and custodial detention broadly defined, including police lock-up, immigration detention, jails, prisons, and transition to community for people being released from these settings.

### Theoretical framework

In this review, we used a framework of abolition feminism, to recognize the intersecting gendered and racist harms of the criminal legal system [8, 9], including the dissolution of families, denial of reproductive care, and increased exposure to sexual violence. Abolition feminism resists reformist logics that would increase investment in carceral systems, such as the creation of health programming within prisons. We also applied the Levesque definition of health service access [10] to frame the results in our discussion. Levesque theorizes access as not only dependent on the physical availability of a service or and its affordability, but also that it be approachable, acceptable, and appropriate. This framing allowed us to contextualize the results and identify opportunities for improvement. The needs of people experiencing criminalization and incarceration are distinct from those of the general population and require services that provide access and accommodation for their specific context.

### Inclusion criteria

This scoping review considered published research including both experimental and quasi-experimental study designs including randomized controlled trials, non-randomized controlled trials, before and after studies and interrupted time-series studies. It considered analytical and descriptive observational studies and qualitative research using various methodologies. Systematic reviews that met the inclusion criteria were also considered and relevant studies included.

### Exclusion criteria

The review excluded non-research and articles not in English, studies conducted among men and boys (people without uteruses), studies focusing exclusively on condom use, and studies about sexually transmitted infections that do not address pregnancy prevention.

### Search strategy

The JBI methods use a three-step comprehensive search strategy. First, the clinical librarian (SM) supported an initial limited search of CINAHL to identify articles on the topic. Second, they used the text words contained in the titles and abstracts of relevant articles, and the index terms used to describe the articles for which we developed a full search strategy in the following databases: CINAHL, APA PsycInfo, Gender Studies, Medline (Ovid), Embase, Sociological Abstracts, and Social Services Abstracts. The search strategy (see S1 Appendix), included all identified keywords and index terms, was adapted for each included database and/or information source. Finally, the reference list of all included sources of evidence was screened for additional studies. The JBI method does not require quality assessment of included studies and this was not performed.

## Study selection

Following the search, all identified citations were collated and uploaded into COVIDENCE and duplicates removed. Following a pilot test, titles and abstracts were screened by two independent reviewers (PPH, MP) for assessment against the inclusion criteria for the review. Potentially relevant sources were retrieved in full and their citation details imported into COVIDENCE. The full text of selected citations was assessed in detail against the inclusion criteria by two independent reviewers (PPH, CH). Reasons for exclusion of sources of evidence at full text were recorded. Any disagreements between the reviewers at each stage of the selection process were resolved through discussion, or with an additional member of the author team. The results of the search and the study inclusion process are presented in a Preferred Reporting Items for Systematic Reviews and Meta-analyses extension for scoping review (PRISMA-ScR) flow diagram [11]. See Fig 1.

## Data extraction

Data were extracted from papers included in the scoping review by two independent reviewers (PPH, CH) using a data extraction tool developed by the reviewers. The data extracted included setting, study design, type of participants (currently incarcerated, under community supervision, released, etc.), sample size, procedures, outcomes of interest, and relevant key findings. The studies were grouped into relevant themes/issues.

# Results

## Study characteristics

The 43 included studies were published between 2001 and 2021. See Table 1 below. The settings included 35 studies set in the US [4, 12–45]; three in Brazil [46–48], two in Canada [2, 5], one in the US and Mexico [49]; one in French Guiana [50]; and one in Uganda [51].

In 34 studies, the population was currently incarcerated [2, 4, 5, 12–31, 33, 35, 41, 43, 44, 45–48, 50]. Two studies included formerly incarcerated people [32, 34]; four studies included prison staff [37–40]; one included people under community supervision [49]; and one included people involved in the criminal legal system broadly defined [36]. Additionally, Erickson et al., (2017) [51] focused on sex workers; we included this study because 27% of respondents reported being formerly incarcerated, and sex work is criminalized in Uganda, the site of the study. All 43 included studies focused on women and/or girls: 34 focused solely on women [2, 4, 5, 12–18, 22, 24–27, 29, 31–33, 36, 41,43, 44, 45–48, 50]; and six on adolescent girls [19–21, 28, 30, 34]. No studies specified if trans or nonbinary people were included. One study focused on African American adolescents [20].

Sample sizes varied from 10 [32] to 1396 [4]. One study [35] analyzed distances between abortion clinics and prisons among 75 state prisons and 20 federal prisons. Twenty-four studies used quantitative methods, including 14 surveys [5, 14, 15, 17–19, 22, 24–26, 29, 30, 37, 45]; four secondary data analyses [21, 31, 40, 48]; one randomized control trial [13]; four prospective studies [4, 40, 41, 43] and one study using geo-localization [35]. Our results also included 14 qualitative studies [2, 12, 20, 23, 28, 32, 34, 38, 39, 46, 47, 49–51], four cross-sectional studies [18, 36, 42, 44], and one mixed-methods study [33].

## Outcomes

The main health outcomes of interest included: contraceptive use (pre, during or post incarceration); rate of abortion; knowledge or attitudes about abortion, contraception, and

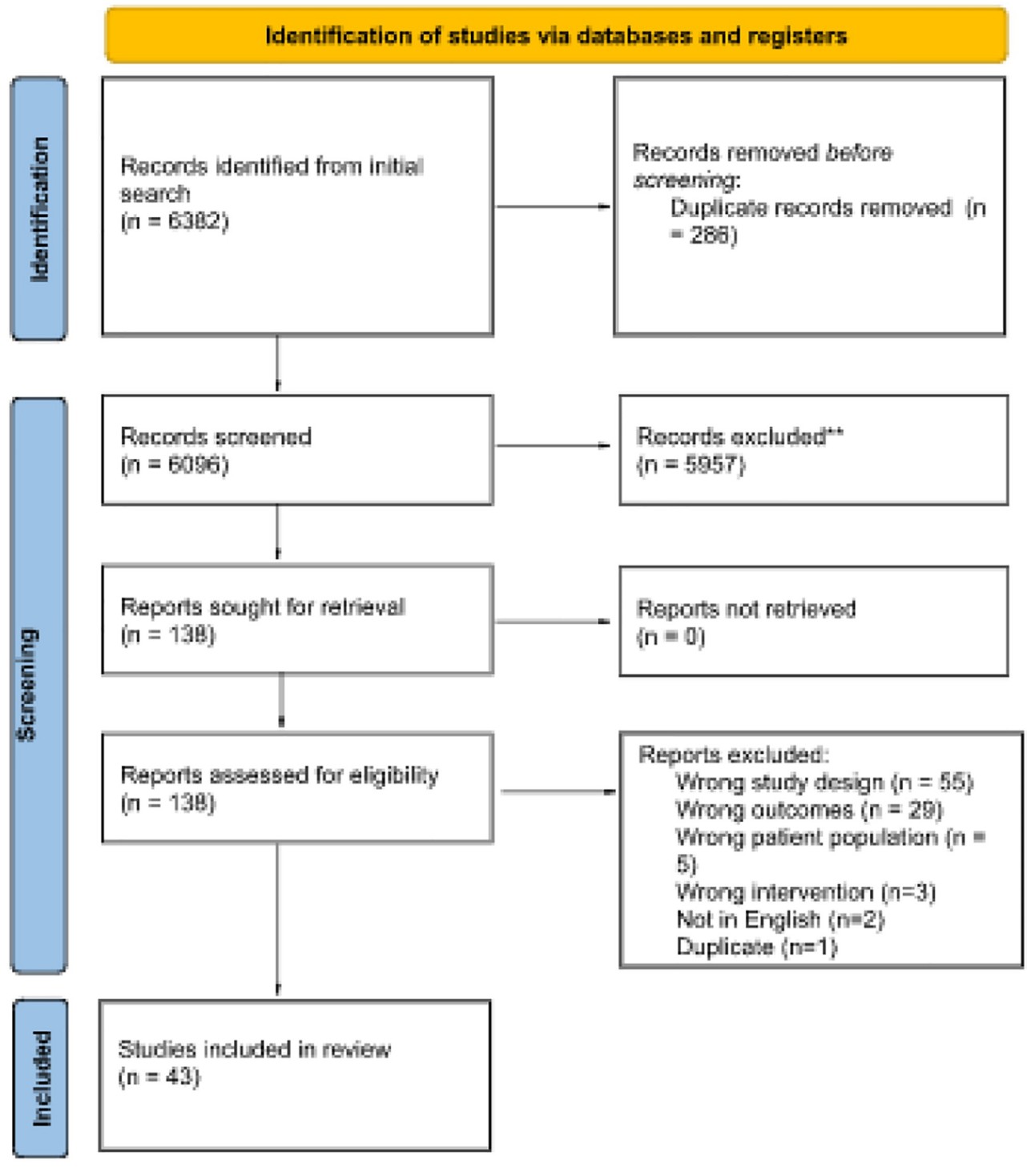

**Fig 1. PRISMA diagram.**

**Table 1. Contraception and abortion access for people in prison.**

| Author and Year | Jurisdiction | Theme/Issue | Aim | Participants | Methods | Outcomes | Results |
|---|---|---|---|---|---|---|---|
| Barros et al. (2016) | Brazil | Rate of abortion | To describe female prisoners' socioeconomic and reproductive profile. | 47 currently incarcerated women | Qualitative (Interviews) | Demographics, reproductive history. | 42.5% of participants had multiple children, 40.4% had no prenatal consultations, and 42.5% had had an abortion. |
| Brousseau et al. (2020) | USA | Contraception use | To assess the efficacy of motivational interviewing as an individualized intervention to increase the initiation of contraceptive methods while incarcerated and continuation after release. | 232 (119 currently incarcerated women; 113 non-incarcerated control group) | Randomized control trial | Initiation of a method of birth control prior to release from the correctional facility, rate of pregnancy, rate of STI, the continuation of contraception | Initiation of contraception was higher in the intervention group. but this difference was not significant after controlling for the number of male partners within the year prior to incarceration. |
| Brousseau et al. (2021) | USA | Contraception use | To understand perceptions of long-acting reversible contraception (LARC) among incarcerated women | 304 (141 currently incarcerated women; 163 non-incarcerated control group) | Cross-sectional survey | Demographics, current and past contraception use, reproductive health care, perceptions of IUDs and implants. | The control population was significantly more likely to use the IUD than the incarcerated population. Condoms were the most common type of past contraceptive method in the incarcerated population. |
| Cannon et al. (2018) | USA | Contraception use | To examine the risk of unintended pregnancy among women during Cook County Jail intake by assessing basic contraceptive history, the need for emergency contraception (EC) at intake, and contraception at release. | 194 currently incarcerated women | Cross-sectional survey | Contraceptive use, pregnancy risk, pregnancy desire. | 17.5% were surgically sterilized or postmenopausal, 4.6% using long-acting reversible contraceptives (LARC), 73.2% at risk for pregnancy. 47.9% of those women had unprotected intercourse within 5 days prior to survey. 81.4% were interested in emergency contraception and 72.7% were interested in contraceptives if provided free at release. |
| Cheedalla et al. (2021) | USA | Institutional policies governing abortion or contraception | To determine contraception policies in incarceration settings around the United States. | 31 administrative staff at state prisons, jails, and juvenile detention centres | Cross-sectional survey | Written birth control policies and procedures, institution demographics, provider of healthcare services, and accreditation by national organizations. | 65% of sites had formal written birth control policies. Policies varied across sites. Sites without policies may still allow birth control; almost all sites (n = 29, 94%) enabled women to initiate at least one method of birth control. |
| Clarke et al. (2006a) | US | Contraception use, Rate of abortion | To assess the pregnancy attitudes and future plans for contraceptive use among a sample of incarcerated women in RI and to identify factors associated with pregnancy attitudes and contraceptive plans. | 223 currently incarcerated women | Structured interviews | Demographics, Substance use history, Sexual and reproductive history, Conception locus of control scale, birth control burden, Want a birth control method now, pregnancy attitudes, Factors influencing decisions regarding contraceptive plans. | About half of the sample endorsed Negative pregnancy attitudes (PA), while 41.3% were categorized as Ambivalent PA and 9.4% Positive PA. The 'Negative PAs' were significantly more likely to have used a birth control method in the past three months. |
| Clarke et al. (2006b) | US | Contraception use | To assess the level of risk for sexually transmitted diseases and the reproductive health needs of 484 incarcerated women in Rhode Island. | 484 currently incarcerated women | Cross-sectional survey | Demographics, substance use history, sexual and reproductive history, birth control history. | 84.4% of participants had ever used a reversible form of birth control, excluding condoms; 69.5% had accessed oral contraceptives. |

*(Continued)*

**Table 1.** (Continued)

| Author and Year | Jurisdiction | Theme/Issue | Aim | Participants | Methods | Outcomes | Results |
|---|---|---|---|---|---|---|---|
| Clarke et al. (2006c) | US | Contraception use | To examined whether incarcerated women would substantially increase birth control initiation if contraceptive services were available within the prison compared with after their release. | 224 currently incarcerated women | Cross-sectional survey | Demographics, substance use history, sexual and reproductive history, birth control history, intention to initiate contraception. | 77.5% of participants reported a desire to initiate use of birth control methods. Within 4 weeks of their release, 4.4% of phase 1 participants initiated use of a contraceptive method. |
| Crosby et al. (2004) | US | Contraception use | To identify the prevalence of health risk factors among a sample of detained adolescent females and determine whether there are racial/ethnic differences. | 197 detained adolescent girls | Cross-sectional survey | Sex-related risk factors | Mean age of first sexual experience was 13 years. The mean number of sex partners was 8.8. 20% tested positive for an STD, 32.2% had ever been pregnant. Of those sexually active 33.9% had not used any form of contraception in the past 2 months. |
| Dasgupta et al. (2017) | US | Rate of abortion | To describe sexual and reproductive health (abortion, miscarriage, contraceptive use, access, and use of reproductive services) of substance-using women involved in the criminal justice system. | 299 women under community supervision and using substances | Cross-sectionallll | Demographic, substance use factors, micro risk environmental factors (physical, social, economic), reproductive health. | 53% reported having an abortion in their lifetime. 46% reported having miscarriages in their lifetime. A larger proportion of women with a history of miscarriage tested positive for STIs (36%) relative to those who had abortions (22%) |
| Deboscker et al. (2021) | French Guiana | Knowledge, attitudes about abortion, contraeeption, pregnancy | To describe incarcerated women in French Guiana experiences in relation to sexual and reproductive health. | 14 currently incarcerated women | Qualitative (Semi-structured interviews) | Reproductive health care, menstrual health, pregnancy-related health, SRH education. | Women considered it useful to have access to contraception in detention. Contraception was also seen as a solution to menstrual cycle disorders. |
| Ely. et al. (2020) | US | Contraception use, Barriers to care | To examine lifetime use rates of various types of contraception, and contraceptive use within the six months prior to incarceration. | 400 currently incarcerated women | Secondary health data analysis | Demographics, Contraceptive use (intention and motivations), substance use | A high percentage of women reported lifetime use of some form of contraceptive; less than one-third of the sample used some form of contraceptive in the last six months prior to their incarceration, |
| Erickson et al. (2017) | Uganda | Rate of abortion | To explore factors associated with lifetime abortions among female sex workers in Northern Uganda | 400 female sex workers | Questionnaires | Demographics, sexual and reproductive health, sex work conditions, HIV, abortion. | Of the 315 FSWs who had been pregnant, 62 (19.7%) had experienced at least one abortion. Lifetime exposure to incarceration retained an independent effect on increased odds of coerced abortion |
| Gips et al. (2020) | US | Distance to abortion care | To determine the proximity of state and federal prisons to the nearest abortion clinic | N/A | Geolocalizationn | Distances between abortion clinics and rural/remote prison facilities. | The farthest minimum distance between a state prison and abortion clinic was 383 miles; the shortest was 2.2 miles. |
| Gray et al. (2016) | US | Contraception use | To examine the social and behavioural factors associated with pregnancy history among a sample of African American adolescent girls recruited from a short-term juvenile detention center. | 188 detained African American adolescents | Interviews | Pregnancy history, individual factors, sexual risk behaviours, pregnancy coercion, contraception. | 58.5% had condomless sex in the past 90 days. Prevalence of pregnancy was 25.5%. |

(*Continued*)

**Table 1.** (Continued)

| Author and Year | Jurisdiction | Theme/Issue | Aim | Participants | Methods | Outcomes | Results |
|---|---|---|---|---|---|---|---|
| Greenwald et al. (2021) | US & Mexico | Contraception use | To examine contraceptive methods used by women on probation and parole and to understand if certain individual and interpersonal factors were associated with method use. | 52 women under community supervision | Cross sectional survey | Pregnancy and contraception attitudes, demographics, current birth control method | 75% of participants reported unintended pregnancies. Permanent contraceptive methods were reported by 34.6% of the participants, and reversible or no contraceptives were reported by 65.4%. High contraceptive self-efficacy was reported by 76.9%. Most participants reported that they made independent decisions about contraceptive choices. |
| Grubb et al. (2018) | US | Contraception use | To assess the effect of providing standardized counselling to improve the rates of contraception initiation and utilization among detained young women | 120 currently incarcerated adolescent girls | Quality Improvement Project | Contraception counselling, initiation, utilization of any contraceptive method. | The QI project showed statistically significant improvements in contraception counselling, initiation, and utilization among adolescents in the detention setting. |
| Hale et al. (2009) | US | Contraception use | To assess the contraceptive needs of women incarcerated in jails | 188 currently incarcerated women | Survey | Birth control use and attitudes, STDs, and pregnancy attitudes. | Although 63.6% of women reported access to a healthcare provider prior to jail, only 25.5% reported access to an obstetrician-gynecologist. **The** most common methods of birth control used in the past were the male condom (74.1%), birth control pills (66.5%), withdrawal (38.9%), or Depo-Provera injection (24.3%), but only 63.5% reported using birth control "almost all the time" during sexual intercourse, and 7% reported no previous use of birth control. |
| Hemberg et al. (2021) | US | Knowledge, attitudes about abortion, contraception | To examined if abortion-related knowledge among women with criminal legal system involvement differed in three U.S. cities in states with varying abortion policies. | 381 women involved in the criminal-legal system | Cross-sectional survey | Abortion-related knowledge between women involved in the criminal system | 45% of participants were placed in the high abortion-related knowledge group. Political differences in the women's regions had an impact on abortion-related knowledge. |
| Kelly et al. (2003) | US | Contraception use, Knowledge, attitudes about abortion, contraception | To develop an intervention targeted to the specific sexual risk behaviours of young women in a juvenile detention center. | 100 adolescent girls in the juvenile detention system | Cross-sectional survey | Sexuality knowledge and attitudes | 94% reported sexual activity, only 58% had used some type of contraception. 91% expressed the belief that "people should use birth control if they do not want a child", 68% indicated they understood they could become pregnant any time during the month. Higher knowledge levels were associated with the consistent use of birth control, talking to partners about sex, and resisting unwanted advances. |

(*Continued*)

**Table 1.** (Continued)

| Author and Year | Jurisdiction | Theme/Issue | Aim | Participants | Methods | Outcomes | Results |
|---|---|---|---|---|---|---|---|
| Kim et al. (2001) | US | Institutional policies governing abortion or contraception | To describe the number of admissions of pregnant adolescents to US juvenile residential systems (JRS) and the outcomes of pregnancies that ended while in custody. | 71 pregnant adolescent girls currently incarcerated in the juvenile residential system | Prospective | Pregnancy-related events: live births, stillbirths, maternal mortality, induced abortions, miscarriages, ectopic pregnancies. | 2/3 JRS allowed abortion during the first or second trimester, 1/3 requested a judge approval. 2/3 cover full or partial costs of the abortion, in 1/3 people had to pay for the procedure. |
| LaRochelle et al. (2012) | US | Barriers to care, Contraception use, Rate of abortion | To describe the utilization of contraceptive services prior to arrest, as well as to assess whether logistical and structural barriers to access of contraception exist among newly arrested women. | 228 currently incarcerated women | Cross-sectional survey | Demographics, attitude towards contraception, reproductive health, barriers to contraception | 21% of women were using reversible contraception at the time of arrest, while 39% had used it within the year prior to arrest. Of the 140 women who did not use reversible contraception in the year prior to arrest, 25 reported that they had wanted to use it during this time. Only 5% reported access to reversible contraception in jail services. |
| Liauw et al. (2021) | Canada | Barriers to care | To explore women's experiences and perspectives of reproductive healthcare in prison. | 21 currently incarcerated women | Qualitative (focus groups) | Attitudes about reproductive health, experiences and attitudes toward pregnancy and contraception, barriers to access contraception. | Women reported limited access to healthcare, limited health personnel and supplies, gender discrimination and limited access to essential reproductive health (contraception and abortion). Participants thought it was important for women to have control over when they get pregnant and wanted access to a gynecologist. |
| Liauw et al. (2016) | Canada | Barriers to care, Contraception use, Rate of abortion | To describe the rates of unintended pregnancy and contraceptive use for incarcerated women in Ontario. | 85 currently incarcerated women | Survey | Attitudes towards contraception, contraception use, unmet need for contraception. | 82% had been pregnant, and of these women, 77% had experienced an unintended pregnancy and 57% reported having undergone a therapeutic abortion. Of women who were at risk for unintended pregnancy prior to incarceration, 80% were not using a reliable form of contraception |
| McNeely et al. (2019) | US | Contraception use | To describe and report pilot data regarding whether the incarcerated women received accurate, comprehensive, and voluntary family planning education and clinical services. | 3678 currently incarcerated women | Program Implementation | Attitudes towards contraception. | Nurses conducted 182 education sessions attended by 3678 women. A total of 921 women requested a LARC. 75% of participants in the sessions had used no birth control during their most recent sexual intercourse before arrest. |
| Miranda et al. (2004) | Brazil | Contraception use | To describe the sociodemographic profile and health problems of inmates in a women's prison | 121 currently incarcerated women | Qualitative (Interviews) | Demographics, clinical and criminal past history. | The mean age of the first sexual intercourse was 15.2 years, 28% reported previous STDs, 9.9% were pregnant at the time of the interview. Most of them reported not using any contraceptive method. |

(*Continued*)

**Table 1.** (Continued)

| Author and Year | Jurisdiction | Theme/Issue | Aim | Participants | Methods | Outcomes | Results |
|---|---|---|---|---|---|---|---|
| Myers et al. (2021) | US | Contraception use | To identify women's contraceptive needs and preferences while incarcerated. | 148 currently incarcerated women | Cross-sectional survey | Attituded towards contraception, contraception use, pregnancy history, demographics. | 73% of participants wanted access to contraception while in jail. 85% of the women reported a previous pregnancy, and 44 participants reported being pregnant while incarcerated (either currently or previously). Participants stressed the importance that contraceptive services can be provided but always be optional and never compulsory or coerced. |
| Pan et al. (2021) | US | Institutional policies governing abortion or contraception | To describe permanent and reversible contraception policies at U.S carceral institutions and the frequency of these procedures. | 28 prisons and jails | Cross-sectional survey | Policy on tubal sterilization for permanent contraception, opportunities to receive postpartum or interval contraception, payment requirements for contraception, availability of on—or off-site reversible contraceptive methods. | 6 prison and 2 jails had written policies about female permanent contraception. 11 prisons and 5 jails allowed patients to obtain permanent contraception. 10 prisons and 6 jails provided access to initiate reversible contraception while patients were in custody. 6 prisons that did not have access to reversible contraception did allow permanent contraception. |
| Ramaswamy et al. 2014 | US | Contraception use | To use both quantitative and qualitative methods to understand factors associated with sterilization use among women leaving a U.S. jail. | 102 recently released women | Mixed methods (Cross-sectional survey and interviews) | Demographics, pregnancy, contraceptive history, and incarceration history. | 32 participants reported having had a tubal ligation. 62 participants reported having an unintended pregnancy. Women who reported sterilization histories had an average of three children (range 1–7), compared to one child for those who did not report a sterilization history. |
| Ramaswamy et al. (2015) | US | Contraception use | To examined factors associated with women's use of highly effective birth control before and after incarceration. | 102 currently incarcerated women | Longitudinal survey | Environmental factors influencing the use of birth control | Close to 90% of women at both time points reported not wanting to be pregnant. Prior to incarceration, 41.7% of women were using highly effective birth control; 33% reported using condoms as their main birth control method; 37% of participants said they would like to initiate a birth control method other than condoms upon release. Post incarceration, 53.7% had initiated or continued a highly effective birth control method since release from jail; 31.5% reported using condoms as their main birth control method. |

(*Continued*)

**Table 1.** (Continued)

| Author and Year | Jurisdiction | Theme/Issue | Aim | Participants | Methods | Outcomes | Results |
|---|---|---|---|---|---|---|---|
| Ribeiro et al. (2013) | Brazil | Contraception use, Rate of abortion | To identify the gynecologic and obstetric profile of imprisoned females in the State of Ceará. | 672 incarcerated women | Health records retrospective study | Demographics, previous abortions, contraceptive methods use. | 31% of the women did not use contraceptive methods at the time of analysis. 45.2% had between 1–2 deliveries, 35.4% 3–4 deliveries, 12.7% 5–6 deliveries and 6.7% 7 or more deliveries. 56.5% never had an abortion, 24.2% had 1 abortion, 16.8% had between 2–3 abortions, 1.1% between 4–5, and 1.4% 6 or more. |
| Saleeby et al. (2019) | US | Barriers to care, Contraception use | To explore attitudes toward contraception and decisions to either prevent pregnancy or become pregnant in girls in the juvenile justice system | 20 currently detained adolescent girls | Qualitative (Interviews) | Demographics, attitudes towards contraception and family planning. | Many participants expressed a strong sense of self-reliance and independence both in their reproductive decision-making and their sexual behaviours. |
| Schonberg et al. (2020) | US | Barriers to care, Contraception use | To understand better the actual contraceptive needs and pregnancy desires experienced by women after incarceration. | 10 women post incarceration | Qualitative (Interviews) | Attitudes towards pregnancy and contraception, preferences for contraceptive services, and access to general and reproductive health care upon re-entry. | Incarceration disrupted women's use of contraception, insurance status and relationship with medical providers. Many women thought that the jail or prison should assist with finding a doctor, making appointments, and providing referrals. Almost all women thought that contraception should have been available at the jail or prison where they were incarcerated. |
| Schonberg et al. (2015) | US | Barriers to care | To understand women's perceptions of receiving contraception at Rikers Island Jail. | 32 currently incarcerated women | Qualitative (interviews) | Experiences with health care and birth control, preferences for contraceptive services, and attitudes toward pregnancy. | The more common barrier to interest in receiving birth control at Rikers was related to perceptions of the quality of medical care offered at the jail. Although almost all participants felt that birth control services should be offered, many stated that they would not use those services themselves. Some thought that taking contraceptives in jail could imply that women were having sex with correctional officers. |
| Sufrin et al. (2009) | US | Institutional policies governing abortion or contraception | To describe current contraception care practices among health care professionals at jails, prisons, and juvenile facilities | 286 health providers at jails, prisons and juvenile facilities | Cross-sectional survey | Policies on Contraception, contraception counselling, birth control prescription. | 19% of respondents reported a formal institutional policy on contraception. 11% were unsure if their facility had a policy, 70% indicated that there was no policy. 71% reported asking women about birth control at some point during incarceration. 55% indicated that women were not allowed to continue birth control method while incarcerated. 38% reported providing women with some birth control method (dispensing or prescribing). Providers identified structural barriers such as administrators who limited their ability to discuss and offer contraception, or lack of time, money, or appropriately educated clinicians. |

*(Continued)*

**Table 1.** (Continued)

| Author and Year | Jurisdiction | Theme/Issue | Aim | Participants | Methods | Outcomes | Results |
|---|---|---|---|---|---|---|---|
| Sufrin et al. (2010) | US | Contraception use, Rate of abortion | To assess the proportion and characteristics of women who would be eligible and willing to take emergency contraception upon incarceration. | 290 newly arrested women | Cross-sectional survey | Demographics, reproductive health history, pregnancy history, contraception use, sexual behaviours. | 69% of participants had delivered a child. 55% had an induced abortion. 32% were using contraception at the time of the survey (13% sterilized, 9% hormonal contraception, 7% IUD, 2% implant 1% barrier method. 85 women were eligible for emergency contraception, 79% of those would not take emergency contraception due to a misperception. |
| Sufrin et al. (2019) | US | Rate of abortion | To collect national data on pregnancy frequencies and outcomes among women in US state and federal prisons. | 1396 currently incarcerated pregnant women | Prospective systematic study | Pregnancy outcomes. | 92% of the sample were live births, 6% miscarriages, 1% abortions, 0.5% stillbirths, 3 newborn deaths and 0 maternal deaths. |
| Sufrin et al. (2021) | US | Institutional policies governing abortion or contraception | To understand abortion incidence among incarcerated people and the relation to prison and jail pregnancy policies. | 22 state prison systems | Cross sectional survey | Abortions, abortion policies in prison systems. | 86% of the prisons allowed abortion, with 58% permitting both first and second-trimester abortions. 3 facilities did not allow abortion under any circumstances. 32% of the prisons did not have a written abortion policy. 14 facilities asked the person to pay for the procedure. |
| Sufrin et al. (2020) | US | Rate of abortion | To describe the number of admissions of pregnant people to U.S. jails and the outcomes of pregnancies that end in custody. | 224 currently incarcerated pregnant women | Prospective systematic study | Pregnancy outcomes | Of the 224 pregnancies that ended in jail, 64% were live births, 18% were miscarriages, 15% were induced abortions, and 1.8% were ectopic. There were two stillbirths, one newborn death, and no maternal deaths. |
| Sufrin et al. (2015) | US | Contraception use | To describe the first five years of experience in providing LARC to women in jail. | 87 currently incarcerated women | Retrospective descriptive study | Demographics, LARC insertion, pregnancy history, history of induced abortion, contraceptive methods use. | A total of 87 LARC devices were inserted in jail during the study period 99% of these women were known to have continued a LARC method for at least one month, including six women who used an implant for the entirety of its recommended duration. Among women with at least 12 months of follow-up information available, 13% of IUD users and 20% of implant users discontinued their method within a year. |
| Thompson et al. (2021) | US | Barriers to care, Contraception use | To explore incarcerated women's perspective of making provider-controlled methods of long-acting reversible contraception (LARC) available in an U.S. urban jail. | 116 currently incarcerated women | Mixed methods (survey and focus groups) | Demographics, attitudes towards contraception and contraceptive use, and attitudes towards LARCs. | 79% of women noted having a regular health care provider prior to arrest with 11% wanting but being unable to obtain reproductive health services in the community in the 6 months prior to their arrest. Participants cited financial concern as the most common reason for not being able to access reproductive health care. |

*(Continued)*

**Table 1.** (Continued)

| Author and Year | Jurisdiction | Theme/Issue | Aim | Participants | Methods | Outcomes | Results |
|---|---|---|---|---|---|---|---|
| Ti et al. (2019) | US | Contraception use | To explore incarcerated girls' experiences of and preferences for family planning care. | 22 currently incarcerated adolescent girls | Qualitative (interviews) | Family planning experiences, attitudes towards family planning, contraception, education and information FP. | 91% reported being sexually active and reported ever used a form of contraception. The most common form of contraception was condoms. 37% of participants were using contraception at the time of the interview. Many girls described feeling stigmatized and judged by providers for not using contraception. Some felt providers were pressuring them to initiate contraception. |
| Wenzel et al. (2021) | US | Contraception use | To examine contraceptive needs among women incarcerated at rural jail. | 95 currently incarcerated women | Cross-sectional survey | Pregnancy history, pregnancy intentions, contraceptive use | 88% of women provided information on birth control use during the 3 months before jail. 37% reported using a birth control method "almost all the time". 37.9% indicated that they would be interested in learning more about birth control methods while in jail. 47.4% indicated that they would be interested in starting or continuing a birth control method while in jail. |

pregnancy; barriers to care; institutional policies governing contraception and abortion; and distance to abortion care.

**Contraception use.** Most of the studies addressed contraception use, including 17 that assessed rates of use of different types of contraception, five that evaluated interventions to improve initiation and use, and six that examined barriers to use.

*Rate of Contraception Use.* Thirteen studies measured use of contraception among adult populations. Brousseau et al. (2021) found incarcerated patients more likely to use condoms than community clinic patients, less likely to use an intrauterine device (IUD) and placed more value on convenience when deciding which method to choose [14]. In their survey of 194 incarcerated women, Cannon et al. (2018) found 4.6% of the sample to be currently pregnant, 17.5% surgically sterilized or postmenopausal, 4.6% using long-acting reversible contraceptives (LARC), and 73.2% at risk for pregnancy due to unmet contraceptive needs [15]. In their interviews with incarcerated women, Miranda et al. (2004) found majority of participants were not using a form of birth control even though they received visits from their sexual partners [46]. Myers et al. (2021) found 44% of their sample had been pregnant while incarcerated, with the most common type of contraception used being condoms, the IUD, and birth control pills [24]. Ramaswamy et al. (2014) found 31% of participants (N = 102) had experienced surgical sterilization [25]. Greenwald et al. (2021) found permanent contraceptive methods were reported by 34.6% of the participants (N = 52), reversible or no contraceptive reported by 65.4% [49]. High contraceptive self-efficacy was reported by 76.9%. Although Ely et al. (2020) found that 96% of their study participants (N = 400) had used contraception in their lifetime, a third had not used any form in the six months prior to their incarceration [31]. Clarke et al. (2006a) found an association between negative attitudes towards pregnancy and likelihood of having used contraception in the past three months [16]. Clarke et al. (2006b) found 84% of

their sample of 484 women had ever used a form of reversible contraception, but 83% had had an unplanned pregnancy [17]. Hale et al. (2009) reported the most common method of birth control used in the past was the male condom (74.1%), followed by birth control pills (66.5%), withdrawal (38.9%), or Depo-Provera injection (24.3%) [22]. Only 63.5% reported using birth control "almost all the time" during sexual intercourse, and 7% reported no previous use of birth control. In Wenzel et al. (2021), the authors found 37% of participants (N = 95) reported using birth control "almost all the time" [29]. Ramaswamy et al. (2015) found prior to incarceration, 42% of study participants used highly effective forms of birth control, compared with 54% after incarceration [26]. Thompson et al. found that just prior to arrest, only 24% of participants (N = 116) were using a non-barrier contraception method, with condoms and withdrawal the most used contraception approaches overall [33].

Four studies measured contraception use among youth. Crosby et al. (2004) found 32% of their sample of 197 adolescent girls in juvenile detention had ever been pregnant, and 34% had not used any form of contraception in the past two months [19]. Gray et al. (2016) found 26% of their sample of 188 girls in juvenile detention had ever been pregnant, and 59% had experienced condomless sex in the past three months [20]. Kelly et al. (2003) found 58% of their sample (N = 100) of adolescents in detention had ever used contraception, and participants had incomplete knowledge about contraception function and efficacy [30]. Ti et al. (2019) interviewed 22 adolescent girls in a juvenile facility and found 91% to be sexually active and 37% reported use of any form of contraception [28]. Echoing these results, Saleeby et al.'s qualitative investigation involving interviews with 20 adolescent girls found six participants had a history of contraception use, with only one using contraception at the time of the study [34]. Participants indicated a desire to prevent pregnancy due to plans and ambitions, but this desire was mediated by a general lack of support and isolation due to incarceration.

*Interventions to Increase Contraception Use*. Five studies evaluated clinical interventions aimed at increasing contraception initiation and continuation of use. Clarke et al. (2006c) found that when incarcerated women were offered a referral to a clinic for contraception prescription after release, only 4.4% initiated use [18]. By contrast, when women were offered contraception while incarcerated, 39% initiated use. McNeely et al. (2019) evaluated a nurse-led family planning information program. Among 3678 women who attended the sessions, 921 requested LARC afterwards, with 794 receiving one (almost all received the implant) [23]. Grubb et al. (2018) measured the impact of a staff and care provider contraception education intervention at a juvenile facility, finding the rate of uptake of contraception (oral pills or injection) increased from 7% at baseline to 52% after the intervention [21]. Brousseau et al. (2020) assessed the efficacy of motivational interviewing to increase contraceptive initiation and continuation among 199 in the intervention group compared to 113 controls [13]. Initiation was higher in the intervention group, 56% vs. 42%, p = 0.03. Sufrin et al. (2015) traced the outcomes of a program to offer LARC to incarcerated women. Of 87 people who had LARC inserted, 53 chose IUDs and 34 chose implants. None of the IUD users experienced complications, and 13% of IUD users and 20% of implant users discontinued device use within 12 months of insertion [27].

*Barriers to Contraception Use*. Six studies discuss barriers to contraception care. Ti et al. (2019) describe youth as having access on-site to only oral contraception and the injection, and travel was required off-site for other options [28]. They felt judged by care providers if they did not use contraception, and coerced to initiate contraception while detained, as one participant described:

"*It made it seem like they really wanted to put me on birth control here. they asked me a bunch of questions, and I answered about like sex and stuff like that. Afterwards they're like*

*trying to get me to start on birth control. And I obviously said, no, because I would rather do that not here, because I'm in here, I kind of just don't want more things that I feel like I have no control over."*

[28, p. 494]

Participants in the study by Myers et al. (2021) reported experiencing coercion from care providers [24].

Participants reported to Ely et al. (2020) that barriers to condom use included not thinking about it or partners not wanting to use condoms [31]. Thompson et al. (2021) found 11% of study participants did not have access to a care provider before incarceration, and many cited financial costs as a further barrier to contraception care. Liauw et al. (2021) found limited access to health care providers and experiences of gender discrimination were barriers to care [2]. Schonberg et al. (2020) conducted interviews with ten women at Rikers Island Jail, who reported that incarceration disrupted their medical are and insurance status, making it difficult to maintain contraception use during incarceration and after release, as one participant illustrated:

*"Well, when I first came home. . .I had to reapply for everything. I had to go to the Medicaid office, fill out all these forms, some of the forms weren't what they needed. I had to come back and it was just like a mess. . ..It took about two months, two and a half months. . ..You know, you just got to wait."*

[32, p.197].

**Abortion access.**   Among the 43 included studies in this review, four examined frequencies of abortion during incarceration, seven measured lifetime experience of abortion, one measured distance to abortion services, two studies addressed participant knowledge about abortion or contraception, and five studies examined institutional policies.

*Rate of abortion*. Only four studies examined frequency of abortion in prison settings. Kim et al. (2021) surveyed three juvenile centres, in which eight pregnancies occurred and one resulted in abortion [41]. Sufrin et al. (2021) found that among the 22 state prisons and 6 county jails surveyed, there were 1,040 pregnancies that ended during the study time period, and 4.2% resulted in abortion [40]. In a large study (n = 1396), Sufrin et al. (2019) found only 1% of pregnancies among people in federal prisons resulted in abortion [4]. In another study focused on county jails, Sufrin et al. (2020) reported that 15% of pregnancies in their sample (n = 224) resulted in abortions [43].

Seven studies reported on lifetime abortion experience. Sufrin et al. (2010) found 55% of a sample of 290 recently arrested women had ever had an abortion [44]. Dasgupta et al. (2017) found that 53% of their sample (n = 299) of women under community supervision had had an abortion [42]. Ribeiro et al. (2013) found that 24.2% of their sample (n = 672) had had one abortion and an additional 16.8% had had between 2 and 3 abortions [48]. Barros et al. 2016 reported that 42.5% of their sample (n = 47) had had an abortion [47]. Liauw et al. (2016) found 57% of their sample (N = 85) had had an abortion [5]. Erickson et al. (2017) found 19.7% of their sample of 400 sex workers had had an abortion [51]. One third of all abortions reported by participants were identified as coerced, and experience of incarceration significantly increased the odds of coerced abortion. LaRochelle et al. (2012) found 54% of their sample of 228 had had an abortion [45]. Clarke et al. (2006b) found 35% of their sample of 484 women had ever had an abortion [17].

*Distance to abortion care.* One study, by Gips et al. (2020), mapped the distance between abortion providers and state and federal prisons in the US. They found that in general, prisons were not located near an abortion provider, with some facilities being located between 2 and 383 miles from care [35].

**Knowledge of abortion or contraception.** Two studies looked at various aspects of knowledge of abortion or contraception. Hemberg et al. (2021) examined abortion knowledge among criminal legal system-involved women and found only 45% of participants had high abortion knowledge. Knowledge was not associated with incarceration [36]. Similarly, Deboscker et al. (2021) interviewed 14 incarcerated women and reported that while participants saw the potential value of contraception to manage challenges with menstruation as well as to avoid pregnancy while in prison, they had a lack of knowledge about sexual and reproductive health [50].

**Institutional policies governing abortion or contraception.** Five studies looked at institutional policies. Three of these, all US-based, focused on contraception access. Cheedalla et al. (2021) surveyed 31 state prisons, jails, and juvenile detention systems in the US and found that 65% of institutions had a written birth control policy [36]. Pan et al. (2021) described contraceptive policies among 28 prisons and jails in the US. They found 28% of institutions had written policies on sterilization (e.g. tubal ligation); 57% allowed sterilization and 57% allowed reversible contraception; and 32% of institutions did not allow reversible contraception but did allow permanent contraception [38]. Sufrin et al. (2009) surveyed 286 health providers in jails and prisons: 70% of respondents reported no contraceptive policy [39]. It is unclear how lack of a policy impacts patient and provider knowledge and actions.

Two US studies examined abortion policies and payment requirements. Sufrin et al. (2021) analyzed the incidence of abortion policies at 22 state prisons and 6 county jails and found policies to be highly restrictive [41]. Half of state prisons allowed abortion in both first and second trimesters, and 14% did not allow abortion at all. Of the prisons that allowed abortion, two-thirds required the patient pay out of pocket. Among county jails, 67% allowed abortion in the first and second trimester, and one quarter required payment out of pocket. Kim et al. (2001) surveyed three juvenile residential centres, finding two allowed first trimester abortion, and one required a judge's approval. Two covered full or partial costs; the other required the pregnant adolescent to pay out of pocket [41].

## Discussion

This scoping review aimed to synthesize what is known about abortion and contraception access with respect to women, trans and nonbinary people experiencing criminalization and incarceration. We learned that incarcerated people are more likely than the public to experience abortion and unplanned pregnancy, and yet face policy, information, stigma, practical and physical distance barriers to contraception and abortion care. These findings begin to fill a knowledge gap about the needs of incarcerated people globally with respect to essential reproductive health services including abortion and contraception.

Abolition feminism recognizes that people experiencing incarceration face not only restrictions on their liberty but also loss of control over their bodily autonomy. They are subject to: high rates of violence, injury, and sexual assault; elevated risks of health harms including infectious and chronic disease; mental health trauma; and limitations in health information and care. Pregnancy in prison is not uncommon: Miranda et al. (2004) found 10% of participants to be pregnant at the time of their study set in Brazil [46]; Myers et al. (2021) found nearly a third of their US-based study participants had been pregnant during their current or a previous experience of incarceration [24]. Further, pregnancy in prison is associated with perinatal

| Availability (physically & timely manner) | Approachability | Affordability | Acceptability | Appropriateness and Adequacy "fit" |
|---|---|---|---|---|
| **Policies for or against/absence of policy** | Coercive, judgmental HCPs | Policies for or against payment | Comprehensive and comprehendible informed consent | Contextual factors (e.g. menstrual management in prison) |
| **Number of HCPs, time HCPs spend in facility** | Correctional staff/policies interference in experiencing health service | Individual prisoner income (e.g. work opportunities, family support) | Cultural and social factors, hyper-incarceration of LGTBQ+, BIPOC people | Lack of access to information |
| **Correctional staff interference in requesting health services** | Distrust of HCPs in punitive institution | | | Lack of peer support |
| **Distance to external services** | | | | |

**Fig 2.**

harms including preterm delivery, low birthweight, and inadequate access to perinatal care [52, 53]. The potential implications of the carceral context on reproductive outcomes are severe and inherently gendered, and people experiencing incarceration must have options to prevent pregnancy and manage unintended pregnancy.

Notably, all but seven of the studies in this review were located in the US. Because the US incarcerates one-third of all incarcerated women in the world [54], it usually dominates research reviews pertaining to the health of women in prisons. There are approximately 231,000 women experiencing incarceration on a given day in the US [55], and three times that many on probation or parole, [56] with significant restrictions on their mobility and activities as conditions of community supervision. Intersecting with these restrictions, bans on abortion access in many states resulting from the spring 2022 Supreme Court decision to repeal Roe v Wade will disproportionately impact criminalized and incarcerated people.

The Levesque definition of access allowed us to consider what access factors the studies address and their limitations. We have conceptualized the manifestations of Levesque access considerations in the carceral environment as outlined in Fig 2 below.

As a measure of availability, Gips (2021) measured distance to services, and several studies examined the presence of policies permitting or forbidding care. Yet none of the studies question the extent to which correctional staff or policies interfere with the very process of receiving care. Is there a requirement, for instance, to submit requests for care through a

correctional officer? Do correctional officers "triage" requests? Is care really available if neither the patient nor the provider govern the pathway to access? Further, having to travel for care may depend on carceral facility escort services, staff availability, and supportive security policy.

Some may argue prison is an opportunity to make health care available to otherwise under-served populations, and international law dictates state responsibility for ensuring health care availability in prison [57]. In this review, Liauw et al. (2021) found participants reported extreme difficulties accessing basic reproductive care [21] and Schonberg et al. (2020) report incarceration to have interrupted participant access to care [32]. Indeed, Canadian research has found, once released, formerly incarcerated people have much higher rates of primary and emergency care use than the general public [58]. None of the studies in our review evaluated the number or type of providers working in the facility, the amount of time they dedicate to reproductive health matters, or their expertise in relation to reproductive health.

Further, availability to people inside a facility will always be subject to the restrictions on availability presented by jurisdictional regulations. Sufrin et al. (2021) found in their study of institutional abortion policies that the most restrictive were in facilities located in states already "hostile" to abortion [40]. Post-Dobbs, this hostility is now more threatening.

Approachability in the carceral context can be jeopardized beyond the usual power differentials between HCPs and patients because of the additional layers of punishment, restraint, and control, and when health care staff are vulnerable to dual loyalty to the institution [59]. Participants in several studies spoke to feeling coerced [24] and judged [28] or feeling mistrust of care provided by prison HCPs (Schonberg et al., 2020). No study addressed the potential impact however of correctional staff or policy interference in the experience of health care. For example, many health encounters would require CO presence and observation, perhaps even by male COs. External appointments could require unclothed body searching on departure and return to the prison.

Affordability can be evaluated with respect to the comparative private cost of a service. Cost is a well-established barrier to contraception, particularly for use of highly effective LARC methods [60]. Many studies identified an unsurprising preference among study participants for low-cost methods such as condoms [14, 22] or oral contraceptives [17], and several addressed policies for or against institutional requirements for private payment. But afford-ability is a factor of the patient's capacity to earn income. While incarcerated, people may be able to work, however prison labour is notoriously poorly paid. Most prisoners rely on family members for economic support. Incarcerated lone mothers may be less likely to receive finan-cial support from partners on the outside [61].

Acceptability refers to the cultural and social factors patients consider. Black, Indigenous and people of colour (BIPOC) and LGTBQ+ community members are hyper-incarcerated: the need for culturally and socially acceptable care is arguably elevated in prisons. For a patient to accept an intervention, the information about the care and approach towards the care must use a comprehensive and comprehensible informed consent process. People experiencing incarceration experience barriers to health literacy [3] that demand greater effort to the con-sent process. In general, this review found that despite the risk of unprotected sex and unin-tended pregnancy being high among incarcerated people [5, 25, 49], contraception use was low. The included studies did not address cultural or social factors that may drive low contra-ception uptake. Only one study specified attention to an underserved and hyper-incarcerated group: African American adolescents [20].

Appropriateness is perhaps the most challenging factor to examine, as assessing the "fit" between patient and intervention is individualized and complex. Few studies evaluated inter-ventions aimed to increase use, such as by providing more information, to test efforts to

improve acceptability. McNeely et al. (2019) found nurse-led education sessions in a facility prompted uptake of LARC [23]. By contrast, in a review of institutional policies, Sufrin et al. (2009) found 55% of responding institutions did not allow women to continue with contraception use once incarcerated [39].

In summary, using the Levesque lens, among these studies we identified there are significant barriers across all categories of access: 1) outright denials of availability, such as policies against or limiting contraception or abortion; 2) inhibitors of affordability, such as requirements for full or partial private payment for care while incarcerated, or costs in community for prescription options; 3) intimidating care providers and contexts that challenge approachability; 4) inappropriate access, such as policies allowing one kind of contraceptive but not another; and finally 5) limited examination of acceptability both in terms of cultural and social factors and the informed consent process,

Looking beyond the access framework to specific populations, the experiences of youth in the studies were particularly concerning. They experienced high rates of sexual activity [30]; one-quarter to one-third of participants had already experienced pregnancy [19, 20, respectively] and between one-third and two-thirds had had unprotected sex in the last two to three months. Considering this context, the findings by Grubb et al. (2018) of increased uptake of contraception among counselled youth could be interpreted as promising [21]. However, this is a population very much in need of non-coercive, non-judgmental, affordable, accessible sexual health information and resources, and detention interferes with consent and compassion.

Finally, one key finding in this review was that rate of abortion, either while incarcerated or during one's lifetime, was the second most common outcome of interest in included studies. Abortion is a frequent experience among the public–one in three people with a uterus will have one in their lifetime [62]. It is more common still among women who experience criminalization; in several studies in this review, the lifetime rate of abortion among participants was over 50% [5, 42]. What remains a question for future research is when abortion occurs in relation to incarceration: are patients seeking abortion in anticipation of pending incarceration, to avoid experiencing pregnancy and birth while in prison? To what extent does unmet contraception need before, during, or after incarceration relate to the elevated rate of lifetime abortion?

Abolition feminism warns against concluding the solution to health service access challenges in carceral context is "more" providers, more resources, or more services *inside* the criminal legal system. The fundamental structures of the systems present incompatibilities with reproductive freedom and autonomous decision-making. Rather, abolition feminism demands health care providers and researchers seek to participate in non-carceral spaces and solutions to improve and expand care, such as community-based clinics and universal coverage for services and prescriptions. The fall of Roe v Wade makes pregnancy management itself now subject to more intense criminalization, and dramatically complicates the risks of reproductive caregiving [63].

## Strengths and limitations

The results of this review should be interpreted in context. The review includes articles written only in English, which limits our understanding of the phenomenon beyond English-speaking countries. The prison sexual health literature includes a large body of research on sexually transmitted and blood borne infection prevention in prisons designated for men, and to tightly focus our review on pregnancy prevention, we did not include these types of studies. As a consequence, our search strategy may have excluded some potentially useful literature. Although trans and nonbinary people are disproportionately subject to criminalization, the prison health literature largely remains binary in its gender-based analyses, and none of the included studies

in this review describe trans or non-binary participants. Strengths of the study include our attention to abortion, too-often avoided in examinations of reproductive health among incarcerated people; international scope; use of abolition feminism as a methodological framework; and use of the Levesque method to organize understandings of barriers to access and remaining gaps in the literature.

## Conclusions

In this review we identified high rates of lifetime pregnancy, unintended pregnancy, and abortion among participants in the included studies, and low rates of contraception use, with particularly low use just prior to incarceration and low use of highly effective but more expensive long-acting, reversible methods. Policies governing access to both contraception and abortion are highly variable, with evidence incarcerated people bear the burden for costs of care privately. English-language research on contraception and abortion access among incarcerated people is highly focused on the US, which has a uniquely high rate of incarceration and increasingly restrictive policies on access to reproductive health services.

Using the Levesque framework, we identify barriers to accessibility in the domains of availability, approachability, affordability, acceptability, and appropriateness. Denial of essential reproductive care such as contraception and abortion in these contexts violates international law. Results of our review indicate there is a critical knowledge gap about abortion and contraception experiences of trans and non-binary incarcerated people and lack of analysis of race-disaggregated data and the role of racism as a barrier to access. How carceral policies and procedures interfere with reproductive health care must be examined. Future research can also explore the interaction between denied access to contraception and abortion and experiences of criminalization, particularly among youth and BIPOC populations, who experience very high needs for nonjudgmental support, information, and care.

## Supporting information

**S1 Appendix. Search strategy.**
(DOCX)

**S1 Checklist. Preferred Reporting Items for Systematic reviews and Meta-Analyses extension for Scoping Reviews (PRISMA-ScR) checklist.**
(DOCX)

## Author Contributions

**Conceptualization:** Martha Paynter, Paula Pinzón Hernández, Clare Heggie, Shelley McKibbon, Sarah Munro.

**Data curation:** Martha Paynter, Paula Pinzón Hernández, Clare Heggie, Shelley McKibbon.

**Formal analysis:** Martha Paynter, Paula Pinzón Hernández, Clare Heggie, Sarah Munro.

**Investigation:** Martha Paynter, Paula Pinzón Hernández.

**Methodology:** Martha Paynter, Paula Pinzón Hernández, Shelley McKibbon.

**Software:** Shelley McKibbon.

**Writing – original draft:** Martha Paynter, Paula Pinzón Hernández, Clare Heggie, Sarah Munro.

**Writing – review & editing:** Martha Paynter, Paula Pinzón Hernández, Clare Heggie, Sarah Munro.

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
