## [Decision Letter · Decision Letter 0]

1 Dec 2022

PONE-D-22-27437Abortion and Contraception for Incarcerated People: A Scoping ReviewPLOS ONE

Dear Dr. Paynter,

Thank you for submitting your manuscript to PLOS ONE. After careful consideration, we feel that it has merit but does not fully meet PLOS ONE’s publication criteria as it currently stands. Therefore, we invite you to submit a revised version of the manuscript that addresses the points raised during the review process. Please address each of the comments provided from the reviewers. I would strongly encourage you to make the suggested revisions that clarify the frameworks that have been applied even if that requires adding additional material to the manuscript.

We look forward to receiving your revised manuscript.

Kind regards,

Andrea Knittel

Academic Editor

PLOS ONE

Journal Requirements:

2. Please complete a PRISMA-ScR checklist (available at https://www.equator-network.org/wp-content/uploads/2018/09/PRISMA-ScR-Fillable-Checklist-1.docx) and upload it as supplementary file.

3. We noted that the database search of your scoping review was completed in 2021. Please ensure that your search is up to date and any relevant studies published since 2021 are included in your scoping review.

6. Please include your tables as part of your main manuscript and remove the individual files. Please note that supplementary tables (should remain/ be uploaded) as separate "supporting information" files.

Reviewers' comments:

Reviewer's Responses to Questions

**Comments to the Author**

1. Is the manuscript technically sound, and do the data support the conclusions?

Reviewer #1: Yes

Reviewer #2: Yes

2. Has the statistical analysis been performed appropriately and rigorously? 

Reviewer #1: N/A

Reviewer #2: N/A

3. Have the authors made all data underlying the findings in their manuscript fully available?

Reviewer #1: Yes

Reviewer #2: Yes

4. Is the manuscript presented in an intelligible fashion and written in standard English?

Reviewer #1: No

Reviewer #2: Yes

5. Review Comments to the Author

Reviewer #1: Abstract:

- Discussion – consider rephrasing statement about judgement as it does not need a full sentence relative to the other things mentioned.

Manuscript

- Introduction – third paragraph could use some clarity proofreading, but explains the gap in the literature well

- Methods – what are the (XX)s?

- Discussion – The inclusion of abolition feminism and the Levesque framework are somewhat clunky. These are important lenses from which to consider the literature, but the discussion of them needs to be better integrated into the review. In the figure, is the left column “Availability” supposed to be bolded? Is it possible to work this in more? Are all of these aspects addressed in studies, or are you identifying them as gaps in the literature? The abbreviation “HCP” is never defined.

There are many grammatical errors throughout. Consider having someone with writing expertise provide editing assistance. Overall, the organization of the manuscript could be improved. Consider focusing on only the Levesque framework or abolition feminism rather than both or reorganizing the manuscript to improve the cohesiveness of these concepts. The introduction of court cases (Roe V Wade and the Dobbs decision,) while useful in a larger context of policy, are not detailed and add an additional concept. There are quite a few moving parts here, and it needs to be simplified.

Table 1: consider grouping the studies by common aims, study designs, or results. Rather than detailing each individual study, focus on reporting findings that support your conclusions. Can you include anything about the Levesque framework in this table so that it is more cohesive?

Reviewer #2: Abortion and Contraception for Incarcerated People: A Scoping Review

Manuscript Number: PONE-D-22-27437

The authors present a study looking at the landscape of contraception and abortion access for individuals who have been incarcerated. Notably, the authors performed a global review of the literature (geographically expansive), and included a reproductively diverse study population (all people who have uteruses).

Introduction

-The authors concisely highlight the need for more information regarding contraception and abortion for incarcerated people

-The introduction references research previously done in the US and changes to US policies regarding abortion access (ie Dobbs), but asserts the importance of this study is the international lens. It may strengthen the introduction to add some information regarding the impact of incarceration on sexual and reproductive health care for international poplations as well

-There is some explanation on why the US predominates the literature in the discussion section, but may be beneficial to address the international need earlier

Methods

No issues

Results

No issues

Discussion

-The authors do an excellent job of discussing the results within the Levesque definition of health service access and with the lens of abolition feminism. While reading, there were a few questions that came to mind that may be of interest for the authors to discuss further?

-Approachability speaks to the relationship between patients/providers. Although only one study (Sufrin, et al) addresses the experience of the providers, it may be interesting to juxatpose the provider barriers with the patient reported barriers to further explore some of the issues further. What makes it hard for HCPs to connect with patients in the carceral setting vs community setting? Did any of the studies explore this?

Highlighting the significance of COs in this dynamic was important. Well done.

-Affordability: This topic can be addressed on an individual and systems level.

Did you find in your review that many institutions were charging patients for contracpetive methods while incarcerated? I didn’t see that in the results. This paragraph seemed to suggest the cost of the method to patients was major influence on contraceptive choice and access while incarcerated. I understand the point being made, but fear it may be misrepresented.

It may be interesting to discuss how the cost of contraceptive methods affect which medications are kept on formulary and/or how this may be an area of further exploration.

This is also a good place to discuss the impact insurance status on access in the community setting to contraception and abortion. Discuss geographical variances.

-The authors appropriately discuss acceptability as an issue among this population. I encourage the authors to explore some of the available literature regarding social determinants of health for incarcerated individuals and consider the interplay on sexual and reproductive health.

-I would add to the discussion the dearth of knowledge around contraception for individuals in the LGBTQ+ community and how the findings in this study can deeply impact their experience both while incarcerated and in the community (especially as they were added to the study population intentionally)

It is discussed in the limitations, but I would consider this a finding and highlight the importance of research geared towards this community

Overall, I thought the authors explored this topic well and used appropriate frameworks to contexutalized the information.

6. PLOS authors have the option to publish the peer review history of their article (what does this mean?). If published, this will include your full peer review and any attached files.

Reviewer #1: **Yes: **Meredith Wise

Reviewer #2: No

---

## [Author Response · Author response to Decision Letter 0]

16 Jan 2023

Responses to Reviewers

Abortion and Contraception for Incarcerated People: A Scoping Review

Manuscript Number: PONE-D-22-27437

Comment Response 

General 

1. Please ensure that your manuscript meets PLOS ONE's style requirements, including those for file naming. Thank you, we have double checked style requirements and updated naming of supporting files to match style requirements. 

2. Please complete a PRISMA-ScR checklist (available at https://www.equator-network.org/wp-content/uploads/2018/09/PRISMA-ScR-Fillable-Checklist-1.docx) and upload it as supplementary file. Attached as supplementary file. 

We noted that the database search of your scoping review was completed in 2021. Please ensure that your search is up to date and any relevant studies published since 2021 are included in your scoping review The search was completed in 2022. Apologies for confusion. 

4. We note that you have indicated that data from this study are available upon request. Our data for the scoping review are the articles cited. 

5. PLOS requires an ORCID iD for the corresponding author Added to corresponding author information and linked in author dashboard. 

6. Please include your tables as part of your main manuscript and remove the individual files. We have embedded into manuscript. 

Reviewer 1 

Abstract:

- Discussion – consider rephrasing statement about judgement as it does not need a full sentence relative to the other things mentioned. Done. 1

Manuscript

- Introduction – third paragraph could use some clarity proofreading, but explains the gap in the literature well Done 2

- Methods – what are the (XX)s? Blinded initials of the authors. These are replaced now with our actual initials. 

 Discussion – The inclusion of abolition feminism and the Levesque framework are somewhat clunky. These are important lenses from which to consider the literature, but the discussion of them needs to be better integrated into the review. In the figure, is the left column “Availability” supposed to be bolded? We have removed this second table and integrated the analysis more deeply. 

Is it possible to work this in more? Are all of these aspects addressed in studies, or are you identifying them as gaps in the literature? Each study does not address all five elements of the Levesque definition. We sought to identify things we did see in the literature and suggestions for further approaches to examining access. 

The abbreviation “HCP” is never defined. Fixed 

There are many grammatical errors throughout. Consider having someone with writing expertise provide editing assistance. We have reviewed the article to improve. Small changes are noted in track changes. 

Overall, the organization of the manuscript could be improved. Consider focusing on only the Levesque framework or abolition feminism rather than both or reorganizing the manuscript to improve the cohesiveness of these concepts. We have aimed to improve the cohesiveness of the concepts 

The introduction of court cases (Roe V Wade and the Dobbs decision,) while useful in a larger context of policy, are not detailed and add an additional concept. There are quite a few moving parts here, and it needs to be simplified. We have removed Roe and Dobbs to simplify. 

Table 1: consider grouping the studies by common aims, study designs, or results. Rather than detailing each individual study, focus on reporting findings that support your conclusions. Can you include anything about the Levesque framework in this table so that it is more cohesive? Thank you for the suggestion to group otherwise, however we are maintaining the alphabetical order. We have reorganized the discussion section about the Levesque framework so that it is less suggestive of a taxonomy or classification of the review studies, and more a holistic consideration of the concepts that appear and the issues that are missing from the literature. 

Reviewer 2 

The introduction references research previously done in the US and changes to US policies regarding abortion access (ie Dobbs), but asserts the importance of this study is the international lens. It may strengthen the introduction to add some information regarding the impact of incarceration on sexual and reproductive health care for international populations as well 

There is some explanation on why the US predominates the literature in the discussion section, but may be beneficial to address the international need earlier

 The other reviewer requested we remove the Dobbs/Roe decisions, which also decreases the US-focus. 

Approachability speaks to the relationship between patients/providers. Although only one study (Sufrin, et al) addresses the experience of the providers, it may be interesting to juxatpose the provider barriers with the patient reported barriers to further explore some of the issues further. What makes it hard for HCPs to connect with patients in the carceral setting vs community setting? Did any of the studies explore this? 

 The Sufrin study among providers examined presence and content of policies, but not qualitative and experiential issues for providers. This is definitely something lacking in the literature, and we have added it to the call for future research. 

Affordability: This topic can be addressed on an individual and systems level. 

Did you find in your review that many institutions were charging patients for contraceptive methods while incarcerated? I didn’t see that in the results. This paragraph seemed to suggest the cost of the method to patients was major influence on contraceptive choice and access while incarcerated. I understand the point being made, but fear it may be misrepresented. The studies did not address who paid for contraception. We have made a few clarifying additions to this paragraph. 

It may be interesting to discuss how the cost of contraceptive methods affect which medications are kept on formulary and/or how this may be an area of further exploration.

 Agree- adding this as an idea for future study 

This is also a good place to discuss the impact insurance status on access in the community setting to contraception and abortion. Discuss geographical variances.

 Yes! We have added this, and an example from our experience in Canada. 

The authors appropriately discuss acceptability as an issue among this population. I encourage the authors to explore some of the available literature regarding social determinants of health for incarcerated individuals and consider the interplay on sexual and reproductive health. 

 We have added more content about structural determinants of health, thank you. 

I would add to the discussion the dearth of knowledge around contraception for individuals in the LGBTQ+ community and how the findings in this study can deeply impact their experience both while incarcerated and in the community (especially as they were added to the study population intentionally) Yes, thank you. 

It is discussed in the limitations, but I would consider this a finding and highlight the importance of research geared towards this community 

 Good point. We have shifted this to the Findings and called for more research.

---

## [Editor Report · Decision Letter 1]

23 Jan 2023

PONE-D-22-27437R1Abortion and Contraception for Incarcerated People: A Scoping ReviewPLOS ONE

Dear Dr. Paynter,

Thank you for submitting your manuscript to PLOS ONE. After careful consideration, we feel that it has merit but does not fully meet PLOS ONE’s publication criteria as it currently stands. Therefore, we invite you to submit a revised version of the manuscript that addresses the points raised during the review process. You have satisfactorily addressed all of the reviewer concerns. I have identified only a few small remaining language issues for your consideration, outlined below.

We look forward to receiving your revised manuscript.

Kind regards,

Andrea Knittel

Academic Editor

PLOS ONE

Journal Requirements:

Additional Editor Comments (if provided):

Thank you for addressing all of the comments from the reviewers. I think the manuscript reads well and meets criteria for publication. I have several small suggestions regarding language in the manuscript that warrant consideration prior to publication.

1. Many authors now use "criminal legal system" instead of "justice system" to reflect how little justice there is in the system. Please consider making this change.

2. Similarly, "carceral facility" or a more specific term such as "detention center" or "jail" is preferred over "correctional facility" due to the paucity of corrections that occur in these facilities.

3. "Unclothed body searches" is preferred to "strip searches."
---

## [Author Response · Author response to Decision Letter 1]

23 Jan 2023

Responses to Reviewers

Abortion and Contraception for Incarcerated People: A Scoping Review

Manuscript Number: PONE-D-22-27437

Comment Response 

1. Many authors now use "criminal legal system" instead of "justice system" to reflect how little justice there is in the system. Please consider making this change. We have made this change throughout the manuscript. 

2. Similarly, "carceral facility" or a more specific term such as "detention center" or "jail" is preferred over "correctional facility" due to the paucity of corrections that occur in these facilities. We have made this change throughout except when referencing “correctional officers/staff” as this is their job title. 

3. "Unclothed body searches" is preferred to "strip searches." Changed, thank you.

---

## [Editor Report · Decision Letter 2]

25 Jan 2023

Abortion and Contraception for Incarcerated People: A Scoping Review

PONE-D-22-27437R2

Dear Dr. Paynter,

We’re pleased to inform you that your manuscript has been judged scientifically suitable for publication and will be formally accepted for publication once it meets all outstanding technical requirements.

Kind regards,

Andrea Knittel

Academic Editor

PLOS ONE
---

## [Editor Report · Acceptance letter]

15 Feb 2023

PONE-D-22-27437R2 

Abortion and Contraception for Incarcerated People: A Scoping Review 

Dear Dr. Paynter:

I'm pleased to inform you that your manuscript has been deemed suitable for publication in PLOS ONE. Congratulations! Your manuscript is now with our production department. 

Kind regards, 

on behalf of

Dr. Andrea Knittel 

Academic Editor

PLOS ONE